# Lasing from Micro- and Nano-Scale Photonic Disordered Structures for Biomedical Applications

**DOI:** 10.3390/nano13172466

**Published:** 2023-08-31

**Authors:** R. Gayathri, C. S. Suchand Sandeep, C. Vijayan, V. M. Murukeshan

**Affiliations:** 1Centre for Optical and Laser Engineering, School of Mechanical and Aerospace Engineering, Nanyang Technological University, 50 Nanyang Avenue, Singapore 639798, Singapore; gayathri.radhakrishn@ntu.edu.sg (R.G.); ssuchand@ntu.edu.sg (C.S.S.S.); 2Department of Physics, Indian Institute of Technology Madras, Chennai 600036, India

**Keywords:** disordered photonics, random lasing, plasmonics, nanomaterials, bioimaging, biosensing

## Abstract

A disordered photonic medium is one in which scatterers are distributed randomly. Light entering such media experiences multiple scattering events, resulting in a “random walk”-like propagation. Micro- and nano-scale structured disordered photonic media offer platforms for enhanced light–matter interaction, and in the presence of an appropriate gain medium, coherence-tunable, quasi-monochromatic lasing emission known as random lasing can be obtained. This paper discusses the fundamental physics of light propagation in micro- and nano-scale disordered structures leading to the random lasing phenomenon and related aspects. It then provides a state-of-the-art review of this topic, with special attention to recent advancements of such random lasers and their potential biomedical imaging and biosensing applications.

## 1. Introduction

The ramifications of light–matter interactions, as well as the quest for a control over these interactions, have opened new pathways to achieve a deeper understanding as well as novel applications in photonics. The presence of imperfections in the medium, specifically, scattering caused by micro- and nano-scale disorders are often treated as detrimental. A deep understanding of these disorders and their interactions with light is crucial for overcoming the limitations in such media and, thus, rendering them useful for applications in optics. Recent studies have revealed a wealth of interesting physics in systems with micro and nano-scale disorders [1]. These findings offer new paradigms of the physical process with potential for yet-unexplored applications in a variety of fields, given that most of the micro- and nano-scale structures around us, from beetle scales to human skin, for example, are disordered in nature.

Scattering centers are distributed randomly in a disordered medium, and light undergoes multiple scattering events in such media. The consequent “random walk”-like light propagation prolongs its path, thereby increasing the number of interactions with the medium that can potentially modify the optical processes involved. Properly harnessing light transport in micro- and nano-scale disordered materials opens up possibilities for light harvesting, sensing, limiting, and other applications [2]. The incorporation of appropriate optical gain mechanisms into such disordered structures gave rise to another exciting field of research called random lasing. The longer interaction time of light in disordered structures in the presence of a gain medium facilitates the amplification of light [3]. Thus, scattering-induced light localization acts as a cavity and provides feedback for lasing under appropriate conditions.

Unlike conventional lasers, random lasers are quasi-monochromatic, coherence-tunable, and multidirectional [4,5,6]. The degree of disorder in the medium primarily determines the emission characteristics of a random laser. This has several implications, one of which is the use of random lasing emission to probe phenomena involving disorder changes in the medium, making it an effective tool for sensing applications [7,8,9]. It also facilitates fundamental research on Anderson localization and other transport phenomena in disordered media [10,11]. Furthermore, the possibility of “mirrorless” cavity lasing leads to miniature laser sources. The complexity of random laser physics, as well as the diversity of materials that are capable of random lasing, motivates more fundamental research in this area. It also provides a scope for engineering random lasers tailored to the desired application. Random lasers have become increasingly popular for biomedical applications in recent years. Our deeper understanding of random laser physics and the swift progress in novel materials and findings in biomedical field highlight the need for an updated and dedicated compilation. This paper, in this context, reviews the recent developments and potential of random lasers generated from micro- and nano-scale disordered media for biomedical imaging and bio-sensing applications. The fundamental science behind the propagation of light in random structures, how it leads to lasing phenomena, and the basic characteristics of such a lasing emission are discussed in the first section. Random lasing has been realized in various types of scatterers, host materials, and configurations to achieve different functionalities for various applications, which are discussed in the second section. The third section reviews recent advances in random-laser-based biomedical applications, particularly bio-imaging and bio-sensing.

## 2. Fundamentals of Random Lasing

A laser consists mainly of three basic elements: a gain medium, a resonator cavity, and a pumping mechanism. The gain medium is a material capable of emitting light when excited using an appropriate pumping mechanism, either optical or electrical. The role of the cavity or resonator is to ensure that the light is trapped within the gain medium for a sufficiently long time for it to become amplified and to surpass the losses imposed by the system, leading to lasing emission. In conventional lasers, the cavity is realized by precisely adjusted optical mirrors to cater energy to certain modes. Generally, the resulting emission is highly coherent, monochromatic, and unidirectional. Scattering is considered to be detrimental in such systems, as it removes photons from the lasing modes. However, it was later found that scattering itself can create a cavity for lasing action, which gave rise to a new class of lasers known as random lasers [12,13,14].

### 2.1. Light Propagation in Random Structures

The inhomogeneity of the medium forces light to change its direction of propagation, leading to scattering. Most of the common materials in our daily lives are disordered structures and owe their appearance to the scattering of light. The strength of scattering depends on the material, its randomness, and the refractive index of its surroundings. The scattering mean free path (*l_s_*), transport mean free path (*l_t_*), and scattering cross-section ($\sigma_{s})$ are the three quantifying parameters used to describe the scattering strength of a material. The scattering mean free path is the average distance that light travels between successive scattering events, whereas the average distance traveled prior to the randomization of its propagation direction is known as the transport mean free path. In other words, *l_t_* is the distance traveled by light before completely forgetting its original direction of propagation. Thus, *l_t_* represents the step size of the diffusion process, and *l_s_* represents the attenuation of coherence. Schematic illustrations of *l_s_* and *l_t_* are given in Figure 1. The relation between these two is given in Equation (1) [15].
(1)$$l_{s}=l_{t}\;\left(1-<\cos\theta >\right)$$
where *θ* is the scattering angle. When $<\cos\theta >$ becomes zero, as in the case of Rayleigh scattering, *l_s_* is equal to *l_t_*. In the case of Mie scattering, $<\cos\theta >$
*≈* 0.5, and *l_s_ ≈ l_t_*/2. The scattering cross section ($\sigma_{s})$ is related to *l_s_* by
(2)$$l_{s}=\frac{1}{n\sigma_{s}}$$
where *n* is the number density of the scattering particles.

Light transport in a disordered medium can be classified broadly into three regimes based on *l_t_* value, with respect to the length of the medium (*L*). If the size of the medium is smaller than *l_t_*, the probability of the photon undergoing scattering before exiting the medium is very low, leading to ballistic transport. However, if the value of *l_t_* is less than the sample size, but greater than the wavelength of light, then the light transport is in a diffusive regime or weak localization regime. If the medium highly scatters with an *l_t_* shorter than the wavelength of light, then it is considered to be under a strong localization regime. The transport regimes are summarized below, and representative diagrams are shown in Figure 2.
(3)$$\mathrm{Ballistic}:\;L\;\leq \;l_{t}$$
(4)$$\mathrm{Diffusive}:\;L\;\gg \;l_{t}\;\gg \;\lambda$$
(5)$$\mathrm{Strong}\;\mathrm{localization}:\;k\;\times \;l_{t}\;\leq \;1$$
where *k* is the propagation constant.

The condition for strong localization, given in Equation (5), is also known as the Ioffe–Regel criterion [16,17]. Light in the strong localization regime is spatially confined similarly to the Anderson localization in electron transport. The transport mean free path is thus a measure of the randomness or the degree of disorder of a medium. It describes the transport mechanism of light within the medium, and can be measured through the coherent backscattering (CBS) experiment.

#### Coherent Backscattering (CBS) Experiment

CBS is an experimental technique used for the estimation of the degree of disorder in a photonic medium [18,19]. Coherent radiation entering a disordered medium undergoes multiple scattering, and the light scattered in the backward direction forms an intensity cone. The full width at half maximum (FWHM) of the cone is inversely proportional to the mean free path, which is a measure of its randomness. The experimental setup used for the CBS measurement is shown in Figure 3. A polarized coherent laser light is used as the probe. The quarter wave plate-analyzer unit blocks photons coming after a single scattering event from reaching the detector. The vertically polarized input laser beam is converted to a right circularly polarized one when it passes through the quarter wave plate. The photons undergoing multiple scattering events preserve this helicity, and those in the backscattered direction are converted back into vertically polarized light by the quarter wave plate, which can pass through the analyzer. However, for single scattering events, a phase difference of π occurs at the scattering interface, which causes a helicity flip, i.e., the right circularly polarized light becomes left circularly polarized light. This is then converted to horizontally polarized light on its return path through the quarter wave plate, and is blocked by the analyzer. In this way, single scattering contributions can be suppressed, and the backscattered light then contains only contributions from multiple scattering events occurring inside the medium. The lens collects the backscattered light and focuses it onto the camera, which records the CBS intensity cone. The FWHM of this cone is inversely proportional to the transport mean free path, which is a measure of the randomness of the medium. The analytic expression for the CBS intensity line shape can be obtained using the diffusion approximation [18]. $I\left(q\right)$, the intensity as a function of the angle θ is given by:(6)$$I\left(q\right)=\frac{3}{16\pi }\;\left[\frac{7}{3}+\frac{1}{{\left(1+ql\right)}^{2}}\;\left(1+\frac{1-e^{-2ql/3}}{ql}\right)\right]$$
where $q=\frac{2\pi \theta }{\lambda }$, θ is the scattering angle, λ is the incident wavelength, and l is the mean free path [18].

### 2.2. Scattering Induced Feedback for Lasing

Light entering a random medium is partially or fully localized depending on the scattering strength, as described earlier. This, in principle, forms a cavity, and in the presence of a gain medium, the scattered light can become amplified. The distance over which the intensity of light is amplified by a factor of *e* is called the gain length (*l_g_*). Lasing occurs when the light is sufficiently amplified to outweigh the losses before escaping the disordered medium. Hence, *l_s_* ≥ *l_g_* is the criterion essential for random lasing to occur [20].

Depending on the strength of localization, there are two kinds of feedback: resonant and non-resonant. If light transport is in the diffusive or weak localization regime, then it supports non-resonant or open-loop feedback. In that case, only a part of the photon energy is returned by the feedback, and the photons may not necessarily reach the initial position. Thus, it lacks spatial resonance, and the resulting emission is incoherent. Also, in the absence of a well-defined cavity, the lifetimes of the photons within the gain volume are independent of their frequency. The emission frequency is determined by the amplification line, which is the only resonant element in the diffusively scattering medium. On the other hand, resonant feedback is found in random media exhibiting strong localization and satisfying the Ioffe–Regel criteria. In such media, light is localized to form closed loops, and these act as resonant cavities for feedback. As a result, sub-nanometer resolution spikes appear in the emission spectra, making them distinct from the spectral profiles of non-resonant random lasing. Random lasers with non-resonant feedback are also called incoherent random lasers, while those with resonant feedback are called coherent random lasers. An illustrative representation of closed-loop/resonant and open-loop/non-resonant feedback is given in Figure 4, along with the typical output spectra resulting from these types of feedback.

The concept of scattering induced feedback was introduced by Letokhov in 1966 to explain the anomalies observed in interstellar emissions [21]. He suggested that scattering due to free electrons and cosmic-dust particles is responsible for the feedback causing the emission. He also proposed the possibility of obtaining lasing action in the powder form of gain materials that are difficult to be made into large, homogenous crystals [22]. Meanwhile, Ambartsumyan et al. demonstrated lasing emission from non-resonant feedback due to diffusive scattering [23]. This was accomplished by the inclusion of a scattering surface in place of one of the mirrors in a Fabry–Perot cavity. Photons in such a cavity suffer scattering and lose their direction, resulting in incoherent random lasing emission.

The theoretical explanation for lasing in diffusive scattering gain media was given by Letokhov in 1968 [3]. He considered a uniform gain media and solved the photon diffusion equation as given below.
(7)$$\frac{\partial U\left(\vec{r},t\right)}{\partial t}=D\Delta U\left(\vec{r},t\right)+\frac{v}{l_{g}}U\left(\vec{r},t\right)$$
where $U\left(\vec{r},t\right)$ is the energy density of photons; *D* is the diffusion coefficient, given as $\frac{vl_{t}}{3}$; Δ is the Laplacian operator; *v* is the velocity of light in the medium; and $l_{g}$ is the gain length. The general eigen solution for the diffusion equation is:(8)$$U\left(\vec{r},t\right)=\underset{n}{\sum }a_{n}\;\psi_{n}\left(\vec{r},t\right)e^{-\left(DB_{n}^{2}-\frac{v}{l_{g}}\;\right)t}$$
where $\psi_{n}$ and $B_{n}$ are the eigenfunctions and eigenvalues, respectively. This is an exponentially decaying equation. However, under the condition $DB_{n}^{2}\leq \frac{v}{l_{g}}$, the energy density begins to increase exponentially with time. This leads to a critical condition for the lasing threshold.
(9)$$DB_{1}^{2}-\frac{v}{l_{g}}\;=\;0$$

*B*_1_ is the lowest eigenvalue, and is inversely proportional to the size of medium (*L*). Thus, the critical volume for lasing threshold is given by [3]:(10)$$L^{3}\approx \;V_{c\;}\approx \;{\left(\frac{l_{t}l_{g}}{3}\right)}^{\frac{3}{2}}$$

In short, if a photon enters a diffusive scattering medium with gain and satisfies Equation (10), there is a high probability that the photon generates multiple photons through stimulated emission before exiting the medium. This photon multiplication gradually increases the photon density and results in lasing emission. As the photon density increases, gain saturation kicks in and *l_g_* increases, limiting the uncontrolled generation of photons.

In 1994, Lawandy et al. experimentally demonstrated random lasing emission via non-resonant feedback with TiO_2_ microparticles dispersed in rhodamine 640 perchlorate dye solution [12]. The term random lasing was introduced later, in 1995 [24]. The first observation of coherent random lasing, enabling discrete lasing modes, was reported in 1999 with semiconductor powders [13]. This discovery provided direct evidence of the existence of recurrent light scattering, a vital component of Anderson’s photon localization model.

### 2.3. Characteristics of Random Lasing

The prevalence of threshold energy/fluence is one of the most important properties of random lasing emission. At low excitation energies, the gain medium emits a broad fluorescence spectrum. With increasing pump energy, the intensity of emission also increases. Beyond a certain energy threshold, fluorescence is overpowered by stimulated emission, leading to intense lasing emission. This can be observed as a change in the slope of the pump energy vs. the emission intensity plot. Simultaneously, there is also a drastic decline in the emission linewidth [25]. The evolution of an emission spectrum from a random laser and its linewidth narrowing are shown in Figure 5a. For non-resonant feedback, the linewidth generally reduces to a few nanometers, and in the case of random lasing with resonant feedback, spike-like narrow spectral features with sub-nanometer linewidths appear on the emission profile. The threshold energy depends on the fluorescence efficiency of the gain medium and the scattering mean free path of the disordered medium. Several strategies have been developed to lower the lasing threshold by using gain media with high fluorescence quantum yields, scatterers with better scattering cross-sections, increased particle density, and high refractive index contrast with the surroundings [26,27].

The fraction of spontaneous radiation contributing to lasing modes is given by the spontaneous emission coefficient, or the *β* factor. In the case of conventional lasers, the *β* factor is typically 10^−8^ for gas lasers, 10^−5^ for commercial semiconductor lasers, and 10^−1^ for microcavity laser systems [28]. The low value of *β* factor in conventional lasers is due to the stringent requirement of directionality, which results in fewer spontaneous radiations contributing to the lasing process. This also translates to highly directional emission in conventional lasers. On the other hand, the *β* factor of random lasers is roughly of the order of 10^−1^ [29]. The mirrorless cavity model of the random laser causes more spontaneous radiation to contribute to lasing irrespective of its direction, resulting in a higher value for the *β* factor, but the emission is not unidirectional, as would be expected from a large-beta laser. This is the consequence of the inherent design of the cavity in random lasers.

**Figure 5 nanomaterials-13-02466-f005:**
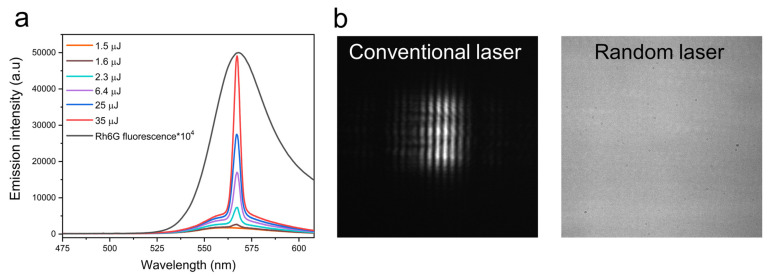
(**a**) The evolution of lasing emission and linewidth narrowing in random lasers. Reproduced from [25] with permission from The Optical Society. (**b**) The far-field interference pattern was obtained by illuminating a double-slit using conventional laser and random laser. Reproduced from [30] with permission from Elsevier.

A high degree of spatial and temporal coherence is another characteristic of conventional lasers. Coherence is a measure of the correlation between electric fields in space and time. The temporal coherence of random lasers has been studied based on photon statistics and interferometric techniques [31,32,33]. First-order temporal coherence accounts for the fluctuations in the field and is related to the emission bandwidth. The narrow bandwidth of random lasers implies high temporal coherence. Second-order temporal coherence accounts for the fluctuations in intensity, and is related to photon statistics. A coherent light source’s photon distribution is Poisson-like, whereas an incoherent light source’s distribution is Bose–Einstein-like. For random lasers with non-resonant feedback, the photon distribution is a superposition of Bose–Einstein and Poisson statistics [31]. However, the single-mode photon distribution for resonant random lasers below the threshold follows Bose–Einstein statistics, and gradually approaches Poisson distribution with increasing pump energies above the threshold [32]. Thus, random laser temporal coherence is a combination of coherent and incoherent components, with the former being dominant at higher pump powers in random lasers with resonant feedback [34].

The spatial coherence of lasers is primarily attributed to its cavity design. The spatial emission profiles and the lasing modes are determined by the cavity’s geometry. In random lasers, the mode competition is significantly reduced due to the absence of a well-defined cavity. As they lack an explicit optic axis, all the randomly propagated modes are augmented to the overall output emission. Altogether, the spatial coherence is drastically reduced [35,36]. The interference fringe pattern obtained from Young’s double slit experiment using conventional laser and random laser illumination is shown in Figure 5b for comparison. The lack of a definite fringe pattern in the case of the random laser clearly shows its low spatial coherence. The spatial coherence is also dependent on the excitation area as well as the scattering strength [37].

Photon degeneracy is another important quantifying parameter for light sources. It is defined as the number of photons in a narrow frequency window per unit of time, area, solid angle, and coherence volume [38]. The value of the photon degeneracy parameter has been found to be of the order of 10^−3^ for thermal sources, 10^−2^ for LEDs, and 10^6^ to 10^9^ for conventional lasers [39]. In the case of random lasers, it ranges from 10^−2^ to 10^3^, with an increasing repetition rate of the pump lasers [39]. Clearly, random lasers have a photon degeneracy value intermediate to that of thermal sources and conventional lasers.

## 3. Random Lasers—Materials and Types

Random lasing has been realized in a wide variety of material systems. Materials with high degrees of disorder or several scattering centers can support the multiple scattering pathways required for random lasing to occur. Additionally, materials with strong light–matter interaction and high emission efficiency can enhance the feedback mechanism essential for stimulated emission. Material properties of interest include a high scattering coefficient, low absorption, and high refractive index contrast. These properties promote efficient light scattering, reduced loss of photons, and effective confinement of light within the gain medium, all contributing to random lasing. Various types of scatterers, host materials, gain media, and configurations have been devised for different applications. This section reviews some of the recent advances in these aspects.

### 3.1. Active and Passive Random Lasers

One of the major categories of random laser uses active scatterers such as semiconductor particles and rare-earth-doped materials, which can provide gain and scattering simultaneously. Semiconductor powders are strongly scattering systems, since their refractive indices are large and the absorption for wavelengths in the bandgap is small. Different morphologies of semiconductor materials such as ZnO, GaN, ZnS, GaAs, etc., have been used to achieve random lasing [40,41,42,43]. ZnO p-n junctions, along with ZnO-based nanocrystallite films and metal-oxide-semiconductor structures, have been demonstrated for their efficacy as platforms for achieving electrically pumped random lasers [44,45,46,47]. Additionally, semiconductors in micropowder and colloidal nanocrystal forms are being explored for random lasing due to their narrow linewidths and adjustable emission properties [48,49]. Rare-earth-element (such as Nd, Er, etc.)-doped powders and fibers have also been reported to exhibit efficient random lasing [50,51,52]. Metal halide perovskites are an emerging class of opto-electronic materials of interest in random lasing due to their tunable bandgaps, high photoluminescence quantum yields, superior light absorption coefficients, and cost-effectiveness [53,54,55]. The presence of scattering centers within the perovskite films, stemming from grain structures, supports the optical feedback in random lasing.

Passive scatterers in combination with various gain media, such as quantum dots, laser dyes, etc., are widely used to obtain random lasing. In the past, random lasers with passive scatterers, such as plasmonic particles, nematic liquid crystals, metamaterials, human tissues, and cellulose structures, have been reported [9,56,57,58]. Disordered photonic structures, such as opals and aerogels, are being explored for their ability to create localized modes and enhanced scattering [59,60]. Liquid crystals offer tunability of the refractive index and birefringence through external stimuli, such as temperature and electric fields, making them adaptable for the purpose of controlling light propagation and scattering for random lasing [61,62,63]. Metamaterials with engineered electromagnetic responses are also garnering attention for their potential to manipulate and enhance light–matter interactions in random lasers [64,65]. Because of the plasmon resonance effect, plasmonic scatterers are very intriguing and constitute a unique class known as plasmonic random lasers. The resonance, particularly localized surface plasmon resonance (LSPR), is more pronounced in nano-scale metal particles and is sensitively dependent on the size and shape of the particle [66]. Thus, metal nanoparticles offer the possibility to tune the emission based on the LSPR effect [67]. Furthermore, localized field enhancement and higher scattering cross-sections provide plasmonic particles with an advantage over their dielectric counterparts, facilitating lower lasing thresholds and narrower linewidths. In addition, the plasmon resonance has been shown to boost the fluorescence of weak emitters by up to 1000 times [68]. This improved lasing performance using plasmonic scatterers was first demonstrated by Dice et al. using silver nanoparticles [57]. Later, Popov et al. also showed that surface plasmons can induce better scattering by using gold nanoparticles in a polymer film [69]. Linewidths as low as 0.5 nm were reported on silver-nanoparticle-based coherent random lasers [70]. Later on, various micro- and nano-scale morphologies of gold and silver particles for random lasing were proposed and demonstrated [71,72,73,74]. Along with plasmonic particles, plasmonic apertures have also been shown to significantly enhance the fluorescence signals. Their responses can be fine-tuned by manipulating the geometry and polarization of incident light, providing dynamic adjustability [75,76].

Ernesto et al. demonstrated the tailoring of scattering cross-sections and the lasing threshold, using dielectric TiO_2_-silica core–shell particles, by tuning the thickness of the silica shell layer [77,78]. Random lasers using plasmonic core–shell particles have also been demonstrated using a similar approach [79,80]. Anisotropic nano-scale plasmonic structures, such as nanorice, nanostars, nanoflowers, and nanowires/rods have also been studied for random lasing [71,74,81,82,83]. Anisotropic scatterers facilitate better spectral overlap with the gain medium while avoiding peak overlap, as they have broader plasmon resonance. This aids in achieving maximum gain coupling with minimal fluorescence quenching and absorption losses [30]. In addition, the broader SPR spectrum of anisotropic structures offers a wide wavelength range over which the scattering properties can be used effectively [71]. Gold nanostars have been explored in this regard for broadband random lasing [71]. Munkhbat et al. recently reported the use of nanostars to create a random laser, whose spectral characteristics could be adjusted by adding multiple plasmonic layers [84]. However, the precise fabrication of anisotropic structures with multiple layers is quite challenging. Structures with sharp or tapered edges, in general, provide electric field enhancement while simultaneously boosting scattering [85]. Figure 6 shows a comparison of electric field localization and hotspot formation in gold nanospheres and nano-urchins [86]. The electric field enhancement of the urchins is two orders of magnitude greater than that of the spheres at their resonant frequencies, and a significant localization of the electric field at the urchin tips can be observed, which results in an intense hotspot region. This enables the random medium to lase, even with low particle density of the order of ∼10^8^ cm^−3^ [87].

The ease of fabrication and size tunability has attracted research attention towards one-dimensional anisotropic scatterers such as nanowires and nanorods. Zhai et al. observed random lasing from Ag nanowires embedded in a flexible substrate at a low threshold of 0.31 MW/cm^2^ [88]. Similarly, a wedge-shaped random laser consisting of Ag nanowires was reported to exhibit lasing at 0.92 MW/cm^2^ [89]. Low lasing thresholds of 0.26 MW/cm^2^ and 0.33 MW/cm^2^ were reported from Ag-Au bimetallic nanowire-based random lasers [90,91]. Wang et al. used nanorods grown using the glancing angle deposition (GLAD) technique for random lasing [64]. Yin et al. reported the dependence of the lasing threshold on the nanorod’s aspect ratio [92]. A systematic investigation using a wider aspect ratio revealed a trade-off between scattering efficiency and LSPR [30]. One-dimensional nanostructures with smaller aspect ratios support LSPR, and those with larger aspect ratios provide better scattering. Hence, the use of scatterers with optimized aspect ratios is necessary in order to achieve efficient and low threshold random lasing [30].

### 3.2. Colloidal and Flexible Random Lasers

The host material for a scatterer-gain system is chosen based on the nature of the materials involved and the purpose of application. The most commonly used host materials are colloids, polymer matrices, fibers, glass-based materials, etc. [93,94,95,96,97]. In colloidal random lasers, the scatterers are dispersed in a suitable gain solution prepared in solvents such as methanol, water, ethylene glycol (EG), dimethyl sulfoxide (DMSO), etc. The choice of solvent depends on factors such as the ability to suspend scatterer particles, the miscibility of the gain material, biocompatibility requirements, etc. In cases in which denser scatterers are involved, viscous solvents like DMSO and EG are used to prevent sedimentation of the scatterers [98]. One of the important advantages of colloidal random lasers is that the amount of scattering and gain can easily be varied by changing the concentration. In the colloidal form, the medium can offer better heat dissipation and, hence, can provide stable lasing output even at higher pump energies. However, the device integration of colloidal random lasers is often challenging.

Polymer-based random lasers are comparatively easier to integrate into device applications. They have found interesting applications due to their flexibility, stretchability, and biocompatibility. The polymer matrix can be an inert host for the scatterer-gain system, and provides material flexibility and processability. In some circumstances, the polymer matrix can provide scattering, gain, or both. Vardeny et al. made an initial observation of coherent random lasing using π- conjugated polymers as the gain media [99]. The structural disorders formed in the polymer during the fabrication process, such as molecular changes, inhomogeneities, and density variations, have been reported to provide scattering feedback for random lasing [95,100,101]. In the case of inert polymers, an appropriate gain medium, usually a laser dye, is doped into them and micro- and nano-scale scatterers are immobilized in the polymer matrix. This is an effective way to avoid sedimentation of the scatterer particles. However, the immobilization of dye molecules has an adverse effect; the population of bleached molecules cannot be replenished by a stream of solution-containing dye as in liquid dye lasers, and this often affects the photostability. Hence, to prevent dye-bleaching, Bhaktha et al. proposed a dye-circulated polymeric microfluidic channel for random lasing [102]. The spectral emission of the polymer random lasers can be tuned by stretching and bending the polymer substrate (see Figure 7) [82,103,104,105]. Random lasers with white emission have also been achieved by combining polymer random lasers emitting red, blue, and green [106,107].

Compared to colloidal (3D) and planar/film (2D) random lasers, the 1D random lasers come with inherent unidirectional emission. This is achieved with the help of optical fibers. In the first demonstration of random fiber lasers in 2007, Matos et al. filled a hollow-core photonic crystal fiber with a suspension of TiO_2_ particles in rhodamine 6G. The fiber geometry is responsible for transverse confinement, whereas the scatterers provide axial feedback in fiber random lasers. Since then, a myriad of materials and approaches have been used in random fiber lasers [10,108,109].

### 3.3. Random Lasers from Biological and Bio-Inspired Structures

Nano-scale architectures in natural photonic structures, such as beetle scales and butterfly wings, exhibit fascinating light-scattering scenarios and interference phenomena [110,111]. In 1990, Yoo et al. demonstrated that biological tissues are disordered photonic structures that can diffusively scatter light and provide weak localization [112]. In 2004, Polson et al. demonstrated random lasing emission from biological tissues for the first time [9]. By impregnating laser dye to tissues extracted from vegetables, animals, and human organs, they were able to show that the random lasing signals from malignant tissues differ from those of healthier ones (see Figure 8a). Further, the tissues can be classified according to the malignancy grade depending upon the lasing spectra, and the power Fourier transform of the spectra gives the cavity length of the resonator in the tissue [113]. Later, random lasing was realized in biological tissues from various parts of the body, such as the brain, bones, blood, etc. [114,115,116].

Biomimetics and bio-inspiration are techniques used to adapt and design efficient structures using the principles of naturally existing systems [118]. Even in the field of random lasers, bio-inspiration has been used as an effective approach for creating efficient scattering media. Leaves are the simplest light-harvesting systems in nature, with intricate surface characteristics that allow for enhanced scattering and effective trapping of light. The papilla structures present in lotus leaves have recently been explored as scatterers in order to achieve mode tunable random lasers [119,120]. The wrinkles and fold structures on monstera and piper sarmentosum leaves have been utilized as natural templates to fabricate PDMS scattering substrates, and interestingly, it was observed that the surface roughness of the structures influences the lasing threshold behavior [121]. Cicada wings doped with gain polymer have been reported to exhibit coherent random lasing [122]. Another interesting study reported the observation of quasi-single-mode random lasing in a butterfly wing skeleton treated with ZnO nanoparticles (see Figure 8b) [117]. Bio-microfibers were recently used to demonstrate a random laser with 33 nm wavelength band tunability [123]. A fully bio-compatible random laser was demonstrated using carrots, in which the cellulose structures act as scatterers and beta-carotene acts as the gain medium [58]. Similarly, an all-marine-based random laser was developed using coral skeletons as scatterers and chlorophyll derived from marine diatoms as the gain material [124]. Research in this direction may open up the paradigm of nature-friendly light sources in the future.

## 4. Biomedical Applications of Random Lasing

Random lasers piqued the attention of various scientific disciplines soon after their first demonstration. Some early notions of the applications of random lasers were to use them as stable optical frequency standards and laser paint [3,125]. Also, they were suggested for studying laser action in substances that cannot be manufactured in the form of homogenous large crystals, which led to the emergence of powder random lasers [22,50]. Furthermore, the mirror-less cavity model has enabled lasing emission at frequency ranges where obtaining high reflecting mirrors is difficult or expensive. Currently, random lasing frequencies range from deep UV to infrared and terahertz [126,127,128,129]. They are also suitable for display applications due to their wide angular distribution over the entire solid angle of 4π. Most importantly, the tunable spatial coherence, multimode behavior, and mode sensitivity with external disturbances make random lasers particularly interesting in biomedical applications such as imaging and bio-sensing [5]. The significant advancements which have been achieved in this direction are discussed in the following sections.

### 4.1. Imaging Applications

Wide-field imaging is one of the most commonly used microscopic imaging technique due to its simple configuration and low cost. It is a prominent technique for real-time, in vivo bio-imaging because of its potential to provide a larger field of view and higher temporal resolution, unlike other imaging techniques where the speed of acquisition is limited by the scanning optics, intricate illumination, and post-processing requirements. In wide-field imaging, the entire field of view is illuminated and imaged, and the image quality is largely dependent on the nature of the illumination source. Lasers are highly desirable sources for imaging applications due to their intense narrowband emission and spectral control. Nevertheless, because of the high spatial coherence, conventional lasers are often incompatible with wide-field imaging. Upon illumination with a highly coherent laser, light scattered from dust particles, the optical surfaces, inherent imperfections in the system, and the sample surface interfere, creating speckles and interference patterns [130]. Such interference patterns and speckles created by the coherent lasers are often known as coherent artefacts. They deteriorate the image quality in wide-field microscopy, making it difficult to interpret the information contained in the images. Several optical and computational techniques have been developed to suppress these coherent artefacts so that lasers can be used for wide-field imaging [130,131]. The most commonly used techniques involve vibrating multimode fibers, scanning micromirrors, and phase randomization techniques [132,133,134,135,136,137,138]. However, all of these methods are sequential decorrelation techniques producing time-varying independent speckle patterns that are to be averaged over many images. For instance, the use of rotating diffusers produces uncorrelated speckles in the images, and the averaging of N such images with independent speckle patterns helps to reduce the speckle contrast by a factor of N^1/2^ [133,135]. A simple estimation revealed that nearly a thousand images need to be captured and processed in order to reduce the speckle contrast to below the human perception level (speckle contrast ratio, C~3%) using this technique [139,140]. Hence, these techniques cannot be used for dynamic imaging or for real-time in vivo wide-field bioimaging applications because of the lengthy acquisition times, vibration noise caused by mechanical movements, and post-processing requirements [131].

Commercial wide-field imaging systems employ spatially incoherent sources such as LEDs, mercury vapor lamps, and xenon arc lamps in combination with excitation filters to avoid coherent artifacts. However, these are broadband sources relying on spontaneous emission and have low photon degeneracy, which implies that even if the source is bright, the number of photons available per mode is sparse. It was in this context that Redding et al. proposed and demonstrated the use of a random laser as an illumination source in wide-field imaging for the first time in 2012 [39]. Low spatial coherence and high photon degeneracy are the two key characteristics that make random lasers appealing for imaging applications. This unique combination is absent in other light sources (e.g., conventional lasers, thermal light sources, or LEDs); see Figure 9 for a comparison of different illumination sources. Figure 10 shows a comparison of the speckle-free imaging capability of a random laser with that of an LED and a conventional laser source [141]. These images clearly show that random lasers generate speckle-free images, unlike conventional laser sources. As described earlier, one of the state-of-the-art techniques used to deal with laser coherence is the use of a dynamic diffuser. The second column of Figure 10 shows images acquired by averaging 15, 100, and 1000 images obtained by employing a conventional laser in combination with a laser speckle reducer (LSR). However, it should be noted that this technique needs at least 1000 images to be averaged in order to achieve an image quality and correlation coefficient comparable to those of random lasers [141]. Previous works have also shown that random lasers perform as well as LEDs, even in scattering environments [39]. The findings suggest that random lasers could be used to image through turbid media. Ma et al. developed a multi-mode random fiber laser in 2019, and to demonstrate its potential for bio imaging, they used a cuvette filled with milk before and after the imaging sample in order to create a bio-scattering environment [142]. In the same year, Lee et al. used a curvature-tunable random laser to exhibit low-noise speckle contrast imaging of dynamic phenomena, such as the blood flow patterns in the ear skin of mice [105]. Random fiber laser has also been used to obtain high-contrast in vitro dental imaging in the backscattering configuration (see Figure 11) [143]. Recently, in 2021, Pramanik et al. developed a portable and low-cost continuous wave random laser for imaging applications [144]. Further, the tunable coherence of random lasers presents unique possibilities for tailoring light sources to specific imaging applications.

Recently, random lasers have been investigated for wide-field fluorescence bio-imaging applications [141,145]. It has been observed that in both trans-illumination and epi-illumination configurations, random lasers outperform LEDs and conventional laser sources and provides high-contrast images with all the finer structural details, as shown in Figure 12. The photon degeneracy and the spectral characteristics of the illumination source have a significant impact on the emission intensity in fluorescence imaging. LEDs have broad emission spectra and low photon degeneracy by nature. Despite the fact that the LED’s bandwidth was limited to 25 nm using bandpass filters (similar to the filters commonly used in fluorescence microscopes), the images (Figure 12a) show that they were still inefficient in comparison to laser sources for the excitation of fluorescent molecules. Conversely, despite having a narrow band width and high photon degeneracy, the images recorded using the conventional laser had lower contrast than the images recorded using the random laser (Figure 12b). This has been explained based on the non-uniform excitation caused by the inherent coherent artefacts in the case of the conventional laser, which led to output distortion, a decrease in overall intensity, and degradation of the image quality. The non-uniform excitation also causes the loss of information that may lead to misinterpretations and errors in quantitative fluorescence imaging. Random laser illumination was also demonstrated to provide better contrast and a higher signal-to-noise ratio in multi-layered diffusive tissue samples [141].

### 4.2. Sensing Applications

The spectral features of random lasers exhibit a strong dependence on the scattering environment. As a result, they display sensitivity to various external factors, including temperature, refractive index, pH, humidity, etc. [7,8,146,147]. This inherent sensitivity allows for a wide range of sensing applications using random lasing [148]. Figure 13a shows the use of random lasing signals for pH sensing. Random lasers that can sense multiple parameters simultaneously have also been developed [149]. A random-laser-based sensor using gold nanoparticles has demonstrated the ability to detect even nanomolar levels of dopamine (a neurotransmitter found in brain tissues) [150]. This has significant implications for the detection and management of Parkinson’s and Huntington’s diseases. Random lasing signals have been instrumental in identifying the morphological and structural alterations triggered by mutations in the Huntingtin gene [151]. In another recent report, a fiber-based plasmonic random laser was utilized to detect human immunoglobulin (IgG) and quantitatively monitor its concentration through specific binding interactions with protein A [8].

Chemical sensors based on random lasers, when operated near or above the threshold, exhibit a sensitivity more than 20 times higher than those relying on spontaneous emission or fluorescence [152]. Further, the use of biocompatible polymers has facilitated random-laser-based wearable sensors that may be transferred or implanted on the skin for the purpose of monitoring physical activities, detecting sweat, etc., as shown in Figure 13b [153,154]. Furthermore, recent research has successfully demonstrated self-healing random lasers which can self-recover their lasing action after being chopped into tens of pieces in only a few minutes [155]. Wearable systems like soft bioimaging implants, portable laser gadgets, and photonic skins could benefit greatly from such random lasers with stretchability, high deformability, and self-healing properties.

**Figure 13 nanomaterials-13-02466-f013:**
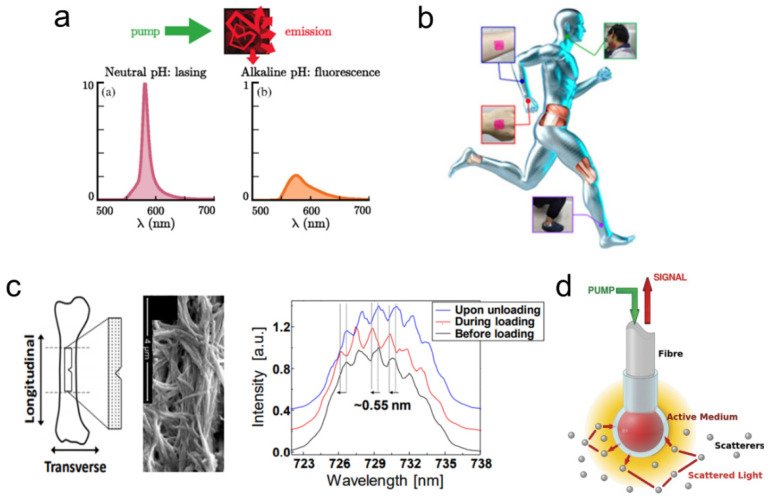
(**a**) Random-laser-based pH sensing. Reproduced from [146] with permission from the American Physical Society. (**b**) Wearable polymer film random laser sensor. Reproduced from [153] with permission from MDPI. (**c**) Random laser for detecting structural changes in bone. Reproduced from [156] with permission from The Optical Society. (**d**) Optical fiber with active medium attached in a spherical cell at its end. Reproduced from [157] with permission from Springer Nature.

As described in Section 3.3, random lasing has been conducted in various biological tissues and is being utilized for biosensing and biodiagnostic applications. Polson et al. used random lasing emission from tissues to identify cancer regions [9]. They observed that the cancerous tissues generated more laser lines than the healthy tissues did. This indicates the presence of more laser resonators as a result of increased tissue disorder associated with cancer progression. The random lasing thresholds were found to be related to the tumor malignancy grade, and could, thus, be utilized to classify tissues for diagnostic purposes [113]. Further, the cavity length of the laser resonators can be estimated by the power Fourier transform of random lasing spectra, which aids in the mapping of cancerous regions [158]. The random-laser-based biosensor developed by Song et al. could detect nanoscale structural and mechanical deformation in the bones (see Figure 13c) [156]. Such mechanical behavior testing with random lasing emission has been extended to soft tissues as well [159]. In these techniques, the tissue acts as the scatterer, and the gain medium is a suitable fluorophore impregnated to the tissues. Recent research has also shown novel strategies for random lasing without the need to infiltrate the biological samples with dyes. Instead, the gain medium is encapsulated in a transparent spherical cell and attached at the ends of an optical fiber, allowing for non-invasive diagnostics of biological samples (see Figure 13d) [157]. Proceeding one step further, a recent study demonstrated that random laser emission can be used to differentiate tissues, for example, fat, nerve, muscle, and skin tissues, even under room light conditions [160].

## 5. Conclusions

The fact that most of the structures in our environment are disordered in nature provides a strong motivation and desire to understand and regulate the manner in which light interacts with these structures. The last two decades of research in this area have greatly advanced our fundamental understanding and have resulted in many practical applications, including random lasers. In this review, the basics of random lasing from micro- and nano-scale photonic disordered structures, scattering feedback mechanisms, and characteristics of random lasing are explained first. The mirrorless cavity of the random laser provides the opportunity to design the lasing cavity according to the intended use, which has been beneficial for many applications, in particular biomedical applications. Additionally, this allows for countless material combinations and possibilities. It is important to select the appropriate scatterer material and a compatible lasing medium for random lasing to be as beneficial as possible for its intended application. In general, film-based or fiber-based random lasers would be preferred for sensing applications due to their directional emission, modifiable environment, and convenience of device integration. For imaging applications, however, lasing from colloidal disordered structures is preferable, because it can withstand high-power pumping with better heat dissipation and can avoid photobleaching of the gain medium, which effectively provides the intense output needed for illumination. The characteristics of random lasing, such as high photon degeneracy and low spatial coherence, make it an excellent source for wide-field fluorescence bio-imaging. In addition to uniform excitation, the narrow bandwidth of random lasing can assist with selective fluorophore excitation in samples containing multiple fluorophores. This can aid in reducing background and bleed-through, allowing for higher-contrast imaging. Furthermore, with the use of random lasers as illumination sources, it is possible to switch more quickly between bright-field imaging and fluorescence modes, which would be of potential interest in in vivo bioimaging and fluorescence-guided surgeries. Recent studies have also pointed to the potential of random lasing for the quantitative assessment of fluorescence emission with high temporal resolution. The real-time frame rate is only constrained by the fluorophore efficiency and the intensity of the random laser source, since no additional image processing or mechanical movements are involved. It is anticipated that the recent developments in high power, tunable, directional, and electrically pumped random lasers will enable the creation of adept wide-field fluorescence imaging systems with high temporal and spatial resolution for in vivo bioimaging in real time. Along with wearable and self-healing photonic devices based on random lasing, the tissue differentiation and nano-scale detection capabilities of random lasing have the ability to take medical diagnostics and treatments to a new level. The short-pulsed nature of random lasing can offer spectral feedback in real time, which can increase safety during laser surgeries by accurately identifying the surgical margins in order to reduce the risk of inadvertent tissue damage, particularly to nerves. With rising artificial intelligence (AI) interventions, spectroscopically controlled automatic laser surgery systems based on random lasing are not far off.

## Figures and Tables

**Figure 1 nanomaterials-13-02466-f001:**
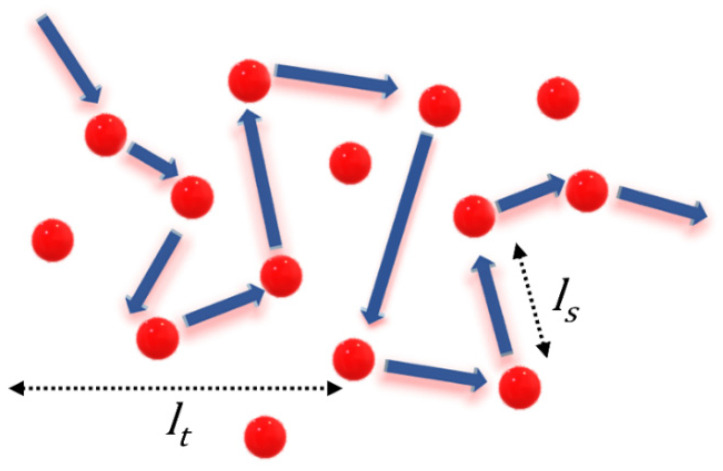
Path of photons in a disordered medium showing a schematic representation of the transport mean free path (*l_t_*) and scattering mean free path (*l_s_*).

**Figure 2 nanomaterials-13-02466-f002:**
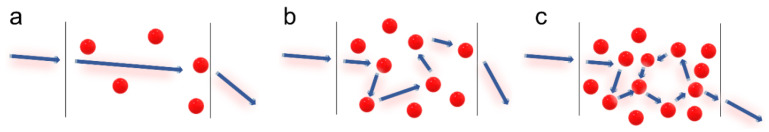
Different light transport regimes: (**a**) ballistic, (**b**) diffusive, and (**c**) localization regime.

**Figure 3 nanomaterials-13-02466-f003:**
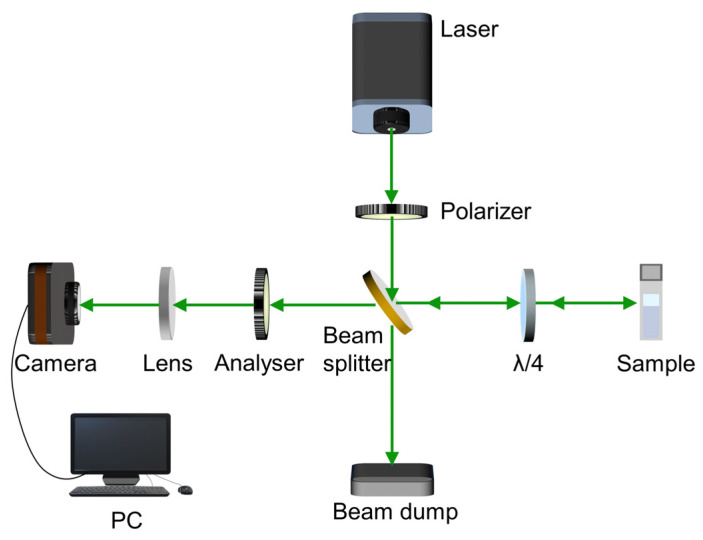
The optical setup for the coherent backscattering experiment.

**Figure 4 nanomaterials-13-02466-f004:**
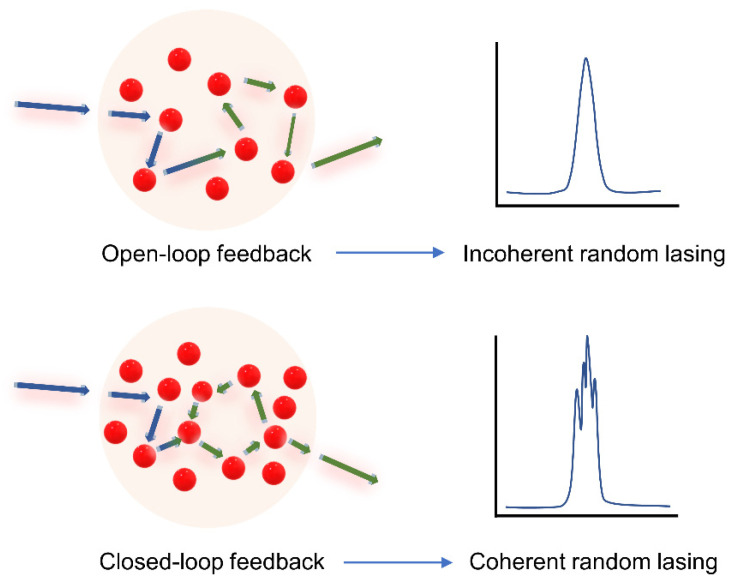
Illustration of non-resonant or open-loop feedback and resonant or closed-loop feedback in random lasers and their corresponding spectral outputs.

**Figure 6 nanomaterials-13-02466-f006:**
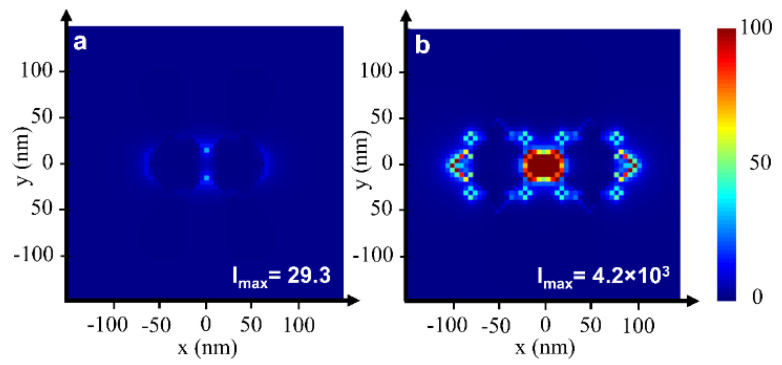
The electric field intensity profile, showing the hotspot region formed between two gold (**a**) nanospheres and (**b**) nano-urchins at their resonant frequencies. Reproduced from [86] with permission from The Optical Society.

**Figure 7 nanomaterials-13-02466-f007:**
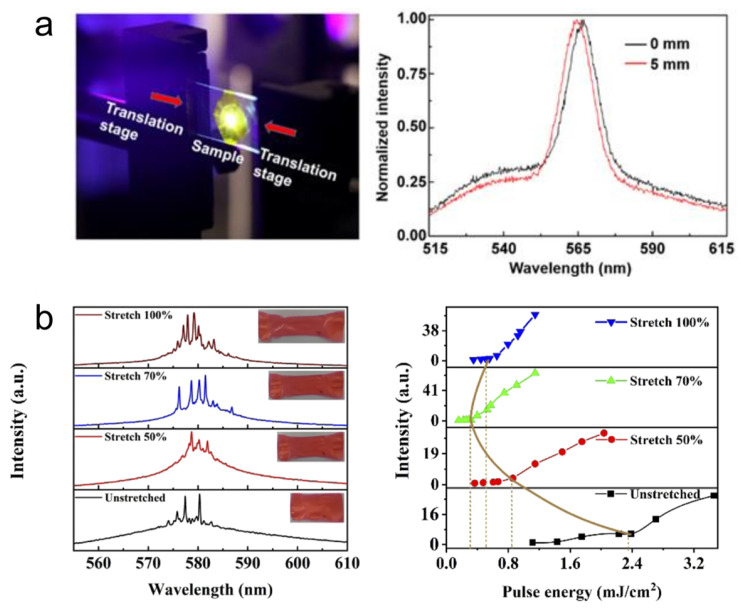
(**a**) Tuning the emission wavelength of polymer random laser by bending. Reproduced from [82] with permission from MDPI. (**b**) Tuning the emission wavelength of polymer random laser by stretching. Reproduced from [103] with permission from The Optical Society.

**Figure 8 nanomaterials-13-02466-f008:**
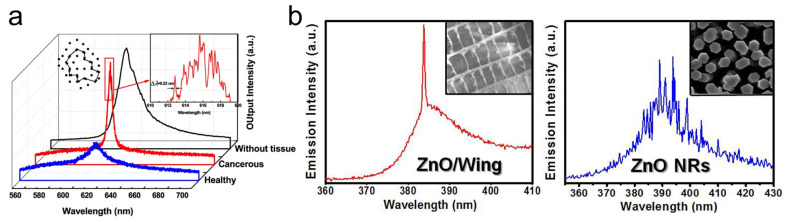
(**a**) Emission spectra from healthy and cancerous breast tissues. Reproduced from [113] with permission from Springer Nature. (**b**) Quasi-single mode random lasing emission in ZnO nanoparticles/butterfly wing composite and multimode spectrum of ZnO nanorods. Reproduced from [117] with permission from Springer Nature.

**Figure 9 nanomaterials-13-02466-f009:**
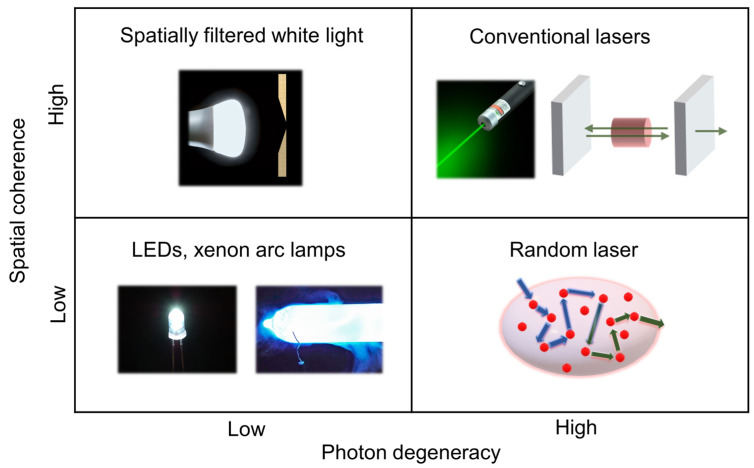
Comparison of light sources based on their spatial coherence and photon degeneracy.

**Figure 10 nanomaterials-13-02466-f010:**
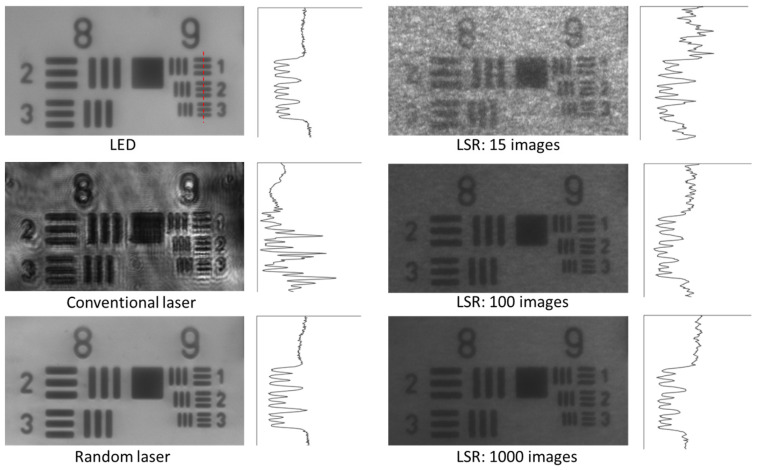
Speckle-free imaging of a 1951 USAF high-resolution test chart using a random laser in comparison to other standard sources of illumination, such as LEDs and conventional lasers. The second column shows the results obtained by illumination using a conventional laser, in conjunction with a laser speckle reducer (LSR), by averaging 15, 100, and 1000 images, respectively. The conventional laser image’s intensity was reduced by a factor of 2.5 for better representation due to the intense, coherent artefacts. All other images are shown as recorded. The graphs shown next to each of the images represent the intensity profiles of the group 9 elements along the red line in the first image. Reproduced from [141] with permission from the Royal Society of Chemistry.

**Figure 11 nanomaterials-13-02466-f011:**
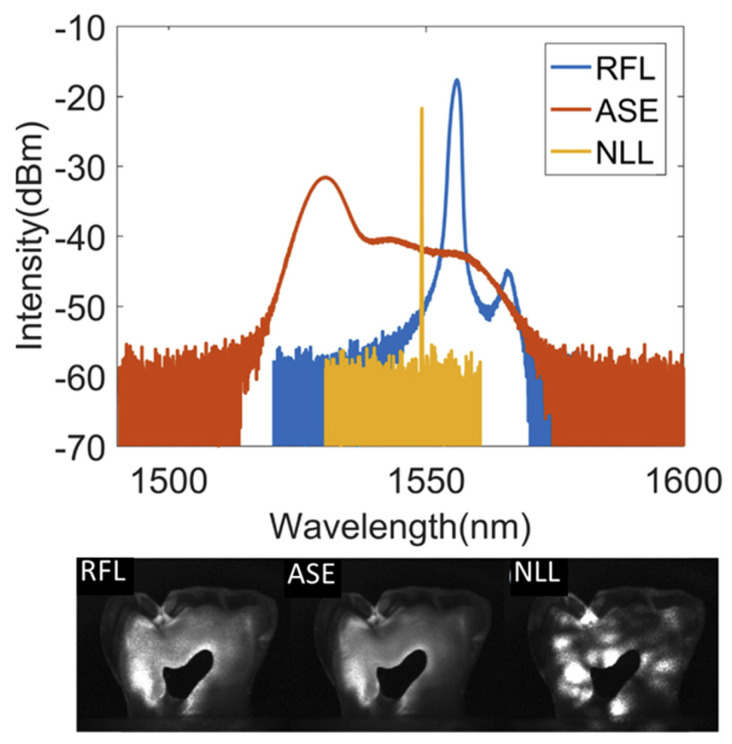
High-contrast dental imaging using random fiber laser (RFL) compared to amplified spontaneous emission (ASE) and narrow linewidth laser (NLL). Reproduced from [143] with permission from The Optical Society.

**Figure 12 nanomaterials-13-02466-f012:**
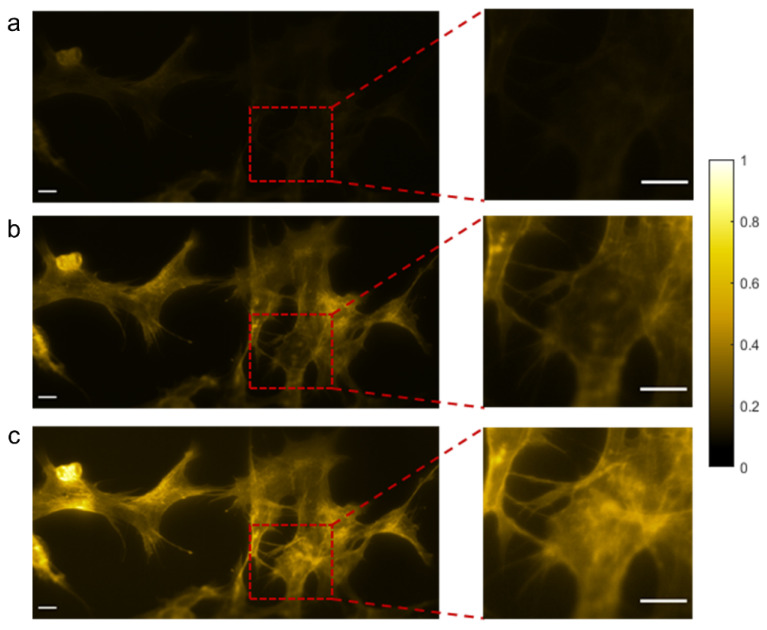
Epi-fluorescence images of human embryonic kidney cells (HEK293T), recorded using three illumination sources: (**a**) LED, (**b**) conventional laser, and (**c**) random laser. The right panel displays enlarged views of the areas in the images that are highlighted in red. Scale bars represent 10 µm. Reproduced from [141] with permission from the Royal Society of Chemistry.

## Data Availability

Not applicable.

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
