# Peer review of "Lasing from Micro- and Nano-Scale Photonic Disordered Structures for Biomedical Applications"

_nanomaterials, 2023, doi:10.3390/nano13172466_

Round 1

Reviewer 1 Report

In the article “Lasing from micro- and nano-scale photonic disordered structures for biomedical applications”, the authors explored the basic principles of physics and associated factors of light movement in disordered structures at the micro and nanoscale, resulting in randomizing events. Following that, we present an up-to-date analysis with a particular focus on recent progress in these random lasers and their possible applications in biomedical imaging and biosensing. Therefore, the reviewer recommend that this article can be published after minor revision in Nanomaterials.

1.     Figures are insufficient in manuscript. Especially, 2.3 Characteristics of random lasing, 3.2. Colloidal and flexible random lasers, and 3.3. Random lasers from biological and bio-inspired structures should have minimal one figure in each chapter for helping the readers’ understanding.

2.     In chapter 4. Biomedical applications of random lasing, the author should show more specific and diverse examples.

3.     It would be good for the author to introduce various examples of electronics production using laser in the last chapter or conclusion. Therefore, I will introduce some reference papers, so please add them.

a.     "BaTiO3-assisted inorganic laser lift-off process for flexible micro-light-emitting diodes." MRS Communications (2023): 1-7.

b.     "Direct freeform laser fabrication of 3D conformable electronics." Advanced Functional Materials 33.1 (2023): 2210084.

c.     "LaserInduced, Green and Biocompatible PaperBased Devices for Circular Electronics." Advanced Functional Materials (2023): 2210422.

Author Response

Please see the replies in the attached pdf file.

Reviewer 2 Report

The paper presents a review of random lasers and its use in biomedical applications. Some modifications are needed before the work can be considered for publication. My comments are listed below.

1. In the introduction section, the motivation of the work is not mentioned. There are exiting review papers on random lasing and their applications. The authors do try to focus on biomedical applications only. However, it is not obvious why the subfield requires a review paper. I would assume the authors do have a proper justification for this. However, it is not made clear in the introduction section of the paper. This is should be corrected in the revised submission.

2. Some key papers on the topic of random lasing and disordered photonics should be cited. For example:

a. https://doi.org/10.3390%2Fs23010247

b. https://doi.org/10.1016/j.nantod.2015.02.006

c. https://doi.org/10.1002/polb.23731

d. https://doi.org/10.1038/s42254-019-0113-8

e. https://doi.org/10.1038/nphys971

f. https://doi.org/10.1117/12.2209623

3. In section 3.1, the paper discusses some of the materials that are used for random lasing. This section requires further elaboration. Mainly, why are certain materials more suitable for random lasing than others? What material properties are of interest? What are the novel materials that are currently being worked on? These details should be discussed.

4. In line 279, it is mentioned that plasmonic structures can boost weak fluorescence signal by 1000 times. This is indeed correct. Along with plasmonic particles, plasmonic apertures and structures have been successfully used to achieve similar effects. In general, plasmonic structures can enhance light intensity and can be controlled through geometry and polarization of the incidents light. A few papers related to this should be cited in this section. For example:

a. https://doi.org/10.1021/nl070822v

b. https://doi.org/10.1007/s11468-022-01735-3

5. In line 460, it is mentioned that random lasers perform as well as LEDs. From line 250, it appears that the photon degeneracy parameter of LEDs is also comparable to random lasers (although a large range is given for random lasers). This begs the question of why random lasers would be preferred to LEDs. Please elaborate this further. Also, do mention why the degeneracy parameter for random lasers span 5 orders of magnitude.

6. Please provide a comparative cost analysis of various light sources that can be used in the same applications as random lasers.

The language is fine. Minor spell checking and rephrasing are required.

Author Response

Please find our replies in the attached pdf file
